# A Comprehensive Review of Reactive Flame Retardants for Polyurethane Materials: Current Development and Future Opportunities in an Environmentally Friendly Direction

**DOI:** 10.3390/ijms25105512

**Published:** 2024-05-18

**Authors:** Paulina Parcheta-Szwindowska, Julia Habaj, Izabela Krzemińska, Janusz Datta

**Affiliations:** Department of Polymer Technology, Faculty of Chemistry, Gdańsk University of Technology, G. Narutowicza St. 11/12, 80-233 Gdańsk, Poland; s180797@student.pg.edu.pl (J.H.); izabela.zagozdzon@pg.edu.pl (I.K.); janusz.datta@pg.edu.pl (J.D.)

**Keywords:** flame retardants, polyurethanes, environmental impact

## Abstract

Polyurethanes are among the most significant types of polymers in development; these materials are used to produce construction products intended for work in various conditions. Nowadays, it is important to develop methods for fire load reduction by using new kinds of additives or monomers containing elements responsible for materials’ fire resistance. Currently, additive antipyrines or reactive flame retardants can be used during polyurethane material processing. The use of additives usually leads to the migration or volatilization of the additive to the surface of the material, which causes the loss of the resistance and aesthetic values of the product. Reactive flame retardants form compounds containing special functional groups that can be chemically bonded with monomers during polymerization, which can prevent volatilization or migration to the surface of the material. In this study, reactive flame retardants are compared. Their impacts on polyurethane flame retardancy, combustion mechanism, and environment are described.

## 1. Introduction

Combustion of various kinds of materials is a complex phenomenon that is dependent on many factors, the most important of which include the following:
Chemical structure and aggregation state of the material;The density, volume, and shape of the product, as well as surface roughness and internal structure;Interaction with the environment, including the dissipation of heat released in exothermic reactions and methods of cooling;Processes of thermal and oxygen degradation and pyrolysis;The influence of heat, radiation, and streams of active particles;Ignition method and time and energy needed to ignite the material under given conditions;Rate of combustion and heat release;The rate of the formation of toxic compounds in the pyrolysis and combustion processes;Ease of extinguishing the fire after it has been ignited [1,2,3,4,5].

Combustion begins with ignition. In practice, the conditions affecting the ignition of polymeric materials vary widely. This is due to the variety of chemical structures, fuel states, ignition sources, and environmental influences. There are two flame mechanisms: global and local exothermic acceleration reactions. The global momentum (self-accelerating mechanism) exothermic chemical reaction occurs in the entire volume occupied by the fuel. It leads to spontaneous ignition of polymeric material, also known as self-ignition. Self-ignition is a spontaneous phenomenon that causes ignition under critical conditions of temperature, pressure, and chemical composition of the combustible material. An explosion is very often the consequence of spontaneous combustion. Exothermic local acceleration reaction of fuel oxidation begins at a specific point in a combustible material under the influence of an external influence. This reaction is associated with forced ignition, which requires the intervention of an external heat source. The created flame spreads freely throughout the entire volume of the system until it reaches its limits.

Various kinds of construction materials have different burning mechanisms depending on their chemical composition, form, and other factors. In addition, the factors influencing the appearance of a fire are also different. The main aim of this article is to describe the burning mechanism of polymer materials, especially polyurethanes, and the possibility of reducing the flammability of these materials. This study provides information about available flame retardants and innovations in the field of flame retardants intended for polyurethane materials. Moreover, the latest research on the synthesis of new flame retardants, the impact of their use in polyurethane materials, and prospects for the development of flame retardants are also described.

## 2. Polymer Material Combustion Mechanisms

Polymers are materials based on hydrocarbons, which can also contain different amounts of oxide and other elements, making them materials with large fire loads. The quick development of polymer materials, including polyurethanes, and their application in many industries, e.g., polymer structural elements, pose a risk of fire related to their working conditions [1,6,7].

The combustion of the polymer usually begins with a forced ignition [4,8,9,10,11]. Under operating conditions, the gas mixture formed as a result of thermo-oxidative decomposition of the polymer most often ignites. The phenomenon of diffusion of reagents has a great influence on the formation of fire. The decisive factor for ignition is the rate of heat accumulation generated in the exothermic oxidation reactions. In the case of positive ignition, the size of the system does not matter. It is necessary to preheat the polymeric material to initiate the oxidation reaction. Low-calorie sources interact with the polymer locally, leading to local heating of the material and creating a combustion center. A fire may occur if there is a possibility of fire spreading.

The ignition of the polymer material is indicated by the appearance of lighting or smoldering [5,12], as well as an increase in temperature in the immediate vicinity. Polymer foam materials, commonly used in many industries, burn very quickly, releasing large amounts of heat, smoke, and toxic gases. Accelerated combustion is favored by the well-developed surface of these materials.

Figure 1 shows the complex overall combustion process of materials. When the ignition sources radiate heat to the surface of the material, volatile flammable products are formed. If the concentration of these compounds is within the flammability limits and also at a temperature above the flash point, the material combustion process continues. Incineration continues as long as the heat applied to the polymer is sufficient to sustain its thermal degradation. Otherwise, the flame will go out. When the heat supply of the ignition source is interrupted or irrelevant, a self-sustaining process occurs for some materials. This process occurs when the heat to support the thermal degradation of the polymer is provided by thermal oxidation either in the gas phase (flame) or in the condensed phase. Processes such as carbonization or condensed-phase thermal oxidation may not occur depending on the type of polymer and combustion conditions [13].

At the polymer–air interface, the smoking process takes place in several zones in which various processes occur. Figure 2 shows the scheme of chemical reactions that take place during polymer combustion in several zones. In the near-surface layer, at the temperature of 400–500 °C, rapid thermal degradation and pyrolysis take place. In the surface layer, oxidation and carbonization of the material surface are carried out. The high-temperature zone is the area above the surface of the material at the very bottom of the flame. In this zone, there is the highest concentration of gaseous combustion products, low-molecular compounds formed as a result of oxidation, and combustion reactions. Temperatures in the flame zone range from 500–800 °C, where the greatest amount of heat is released as a result of burning low-molecular gas products. There is a zone above the flame where products after combustion accumulate, i.e., particles such as CO, CO_2_, -CHO, -OH, etc.

## 3. The Possibility of Reducing the Flammability of Materials

In order to increase the combustion resistance of polymeric material, flame retardants (usually chlorine and phosphorus compounds) are added to the polymer. These agents introduced into the polymer matrix should meet the following conditions:Reduce the overall flammability, as well as the flammability and surface flammability of the polymer;Reduce (or at least not increase) the amount of smoke produced;Not increase the toxicity of gaseous products of combustion;Not change the appearance and affect the functional properties of polymers as little as possible;Increase the cost of the material to a only small extent;Not worsen working conditions during processing;Remain in the plastic material during long-term use.

There are a number of general methods for limiting the flammability of polymeric materials including the following examples:Dilution of the reaction mixture with non-flammable agents;Addition of flame retardants without their chemical bond with the polymer (additions);Addition of flame retardants that react with foam components (reactive agents);Production of chemical bonds resistant to high temperatures during the foaming process;The use of flame-retardant aromatic polyisocyanates in the production of foams;A combination of the above methods;Increase the relative content of aromatic rings;Increase the degree of cross-linking of the polymer.

Nowadays, it is important to develop possibilities for fire load reduction with the use of new kinds of additives or monomers containing elements responsible for material fire resistance [12,15,16,17].

The first method is the addition of materials that are incorporated into the polymer during its processing. For example, additive antipyrines can be added during the processing of the material [5,10,17,18]. Antipyrines do not combine chemically with the molecular structure of the polymer and they can often serve both as plasticizers and fillers. Their cost is lower than that of reactive flame retardants, but their use may lead to an unfavorable change in the performance properties of the material. The most popular additive flame retardants are carbon nanotubes [18], expanded graphite [19], and functionalized graphene oxide [20,21,22].

Mechanisms of additive flame retardants include the following [12,13]:By cooling: the lowering of the temperature of the material via the endothermic reactions that occur in the flame retardant added to the polymer;Owing to the protective coating: the use of compounds that form a protective barrier (carbon) on the surface of the polymer that protect against the penetration of heat into the polymer and reduce gas exchange, thus reducing the flammability of polymers;By dilution: the use of substances that decompose endothermally upon heating by giving off non-flammable gases (water and CO_2_) that physically protect the polymers by reducing the amount of flammable gases.

The additive method of limiting material flammability often requires the use of additional flame retardants in order to obtain a synergistic effect. Moreover, using additives as flame retardants can lead to migration or volatilization of the additive to the surface of the material, which causes a loss of aesthetic product value. Their addition in increased amounts could reduce the mechanical properties of polyurethanes [23,24,25]. Furthermore, low-volatile chemical additives used as combustion inhibitors can also be volatilized, causing a toxic effect on the environment. 

Reactive flame retardants form compounds containing special functional groups, which can be chemically bonded with monomers during polymerization. The use of flame retardants chemically incorporated into the polymer chain prevents volatilization or migration to the surface of the material [5,26]. Reactive flame retardants show higher thermal stability compared to additives due to their chemical bonding with polymer chains. The main advantage of using reactive-type flame retardants is that the polymers are endowed with a permanent flame-retarding performance while retaining their original physical properties [27]. The most popular reactive flame retardants are metal compounds [28,29,30], melamine and its derivatives [31], boron compounds [32], phosphorous flame retardants [6], and polyhedral oligomeric silsesquioxanes [33].

Generally, the use of reactive flame retardants is related to the design and modification of polyols with different chemical structures. Polyols are mostly organic compounds containing flame retardant parts and reactive groups, which can form bonds with other substrates in polymers, thus resulting in improved compatibility [34]. Among the most popular reactive flame retardants used in the synthesis of polyurethanes are phosphorus, silicon, and nitrogen-containing monomers. These atoms are often present simultaneously in one compound to obtain an even better flame retardancy. There are also many reports in the literature about research on the synergistic effect between additive and reactive types of monomers to improve the fire safety of materials.

## 4. Polyurethane Materials

Polyurethanes are obtained by the polyaddition reaction of isocyanates with compounds containing reactive hydrogen atoms such as alcohols and amines. Polyurethanes are an important group of polymers that mainly include foams, elastomers, resins, coatings, and adhesives [35]. These types of materials, consisting of different microstructure variations, are used in many industry sectors such as the footwear market, high-performance adhesives, textile industry, sports equipment, construction, automotive industry, insulation panels, biomedical applications, electrical insulation, etc. [23,36].

Flame retardant-free polyurethanes are flammable materials. This mainly applies to foam products in which the highly developed pore surface facilitates combustion. Polyurethane decomposition usually begins at temperatures above 200 °C. During PUR combustion, toxic fumes and gases are produced. The share of hydrogen cyanide and carbon oxides in the destruction products increases with the rise in temperature. Among the decomposition products of PUR, there are also nitrogen oxides and nitriles. Isocyanate vapors floating above the surface of the burned polymer condense, while liquid polyols further decompose [37,38,39].

The characteristics of elastomers include a large modulus of longitudinal elasticity and tearing module, as well as resistance to scratching, abrasion, weathering, and aging. They can be divided into cast, rolled, and thermoplastic materials based on the processing method [35]. Nowadays, special attention is given to polyurethane elastomers with specific properties such as reduced flammability, high mechanical and thermal stability, or biodegradability [40]. The use of flame retardants during PU elastomer preparation usually leads to selected property modification, e.g., reduction in mechanical properties [41,42,43].

Polyurethane foams can be basically divided into rigid and flexible materials. Rigid foams are characterized by good dimensional stability, low density, high strength, strong aging resistance, and low thermal conductivity [44]. Polyurethane foams have a high surface area to mass ratio and good air permeability, which makes them easily flammable. The fire resistance of these materials is very important to improve safety, primarily in thermal insulations, mattresses, building constructions, and the car industry [24,25]. Rigid polyurethane foams have a higher flash point than flexible foams. The higher content of carbonyl groups, which increases the cohesive energy, makes polyurethane foams less flammable than polyethylene foams. The flammability of PUR foams depends on their type, degree of cross-linking, and the physical condition of the product [17,45,46,47,48].

Another group of materials is polyurethane coatings. Water-based systems are dominating the coating industry as a consequence of their lower toxicity compared to solvent-based products. The main applications of waterborne polyurethanes (WPUs) are leather, wood, and fabric coatings, which indeed are combustible materials. For this reason, modification of WPUs is necessary to prevent fire-related casualties and property losses [7,21,49,50,51].

Polyurethane adhesives are generally used in automotive, furniture, electronics, footwear, and leather finishing. Other potential applications have been limited because of the high flammability of these materials. Proper selection of flame retardants is necessary to prevent the deterioration of physical and mechanical properties and the worsening of the hydrolysis resistance [52].

Currently, the majority of commercially available polyurethane materials are obtained from petrochemicals. The turn of the 19th and 20th centuries marked the beginning of the development of eco-friendly polyurethanes, which can essentially be divided into materials prepared from renewable sources such as vegetable oils or recycling products. They are used in a wide variety of applications including coatings and foams. However, polyurethanes are highly flammable and their further development depends on their flame resistance [23,53].

Polyurethanes in their basic form (without flame retardant additives) are flammable but will not ignite spontaneously at room temperature. CO, CO_2_, and many toxic gases, such as CO and HCN, are released during polyurethane combustion [38,54]. Polyurethane begins to pyrolyze from 200 °C. In the first step, formate groups decompose into isocyanate segments and polyols. Then, the polyols decompose into ethers and alcohols with increasing temperature. From 300 to 500 °C, the residues decompose into amines, ethers, volatiles, and CO_2_. Oxygen has a significant influence on the pyrolysis and combustion of polyurethane. The distribution of the pyrolysis products of polyurethane is determined by temperature, and the primary and secondary pyrolysis products are generated by the breaking of urethane bonds and hydrogen conversion of polyhydric alcohol. Many studies have shown that there are big differences between the pyrolysis and combustion of multiple-system and single-system materials. During the spontaneous combustion process of the coal–polyurethane binary system, there are synergistic effects, which greatly change the mechanism and gas release laws of spontaneous combustion [55]. 

## 5. Phosphorus-Containing Reactive Monomers

Phosphorus-containing compounds have been considered to be the most promising flame retardants (FRs) because of their high flame retardant efficiency and relatively low production of toxic gases and smoke [7]. Organophosphorus compounds have a positive effect on the char formation process, making them very effective. They produce the stable form of poly(meta-phosphoric acid) during burning, which forms an insulating layer on the surface of the material [45,56].

Phosphorous-based flame retardants comprise a wide range of products. Among them, there are red phosphorus (RP) [57,58,59], inorganic phosphates, and organic–inorganic compounds containing incorporated phosphorus atoms [60]. However, red phosphorous has a major disadvantage, which is the release of highly toxic phosphine during melting and requires microencapsulation (mRP) [58,61,62]. Regarding ammonium polyphosphate (APP), there are some problems with its practical use [63,64,65].

Phosphorus flame retardants work in condensed or gas phases. They contribute to the formation of a char yield on the surface of the solid phase, which limits further access of heat and air to the burning sample materials. Phosphorus flame retardants also prevent material decomposition gas products from entering into a flame zone. As a result of the thermal decomposition of phosphorus compounds, most of them form phosphoric acid, which condenses quickly to form pyrophosphate structures (P-O-P) and H_2_O [66]. Phosphorus protects the polymer when the water content is diluted, which rarefies the oxidizing gas phase. Phosphorus-based combustion inhibitors can also take the role of free radical scavengers in a gas phase. When they are incorporated into the polymer structure, they play an effective role in reducing heat release [17,18]. A common phenomenon is the acceleration of the initial degradation of polyurethanes containing incorporated phosphorus atoms at low temperatures due to the lower thermal stability of the phosphorus segments. The energy of the P-O bond (149 kJ/mol) is much lower than the C-O bond (256 kJ/mol) in the main chain of polyurethane [6,27]. Subsequently, the stabilization of the second step of degradation takes place by producing char, which inhibits this process.

S. Bhoyate et al. [24] synthesized a reactive flame-retarded polyol based on phenyl phosphoric acid and propylene oxide (PPA-PO) and used it with limonene-based polyol for the preparation of flame-retarded rigid polyurethane foams. The obtained materials contained 0 to 2% of phosphorus concentration. For the sample, which had the highest amount of PPA-PO, compression strength rose above 100 kPa compared to the reference material. As the amount of phosphorus increased, a smaller flame spread range was observed during horizontal burning tests. The pHRR and THR values for 1.5% phosphorus-containing PU were reduced to 68.6% and 23.44%, respectively, compared to reference foam. Raman spectroscopy results showed that with increasing P content in foam, the amount of charcoal also increased, which caused a reduction in the burning time and mass loss. This means that PPA-PO promoted flame retardancy in the condensed phase. The authors concluded that phosphorus-containing polyol can be mixed with bio-based polyol given flame-retarded rigid PU foams with improved physical–mechanical properties. Moreover, phosphorus-containing polyols incorporated into the foams cause persistent flame retardancy with a homogeneous distribution of flame retardant.

W. H. Rao et al. [25] obtained phosphorus polyol (PDEO) during the synthesis of phenyl-phosphoric dichloride and ethylene glycol, which they then used to make flexible polyurethane foams with the presence of polyether polyols and toluene diisocyanate. They obtained materials containing 0, 5, 10, and 15 php of PDEO. It was observed that the incorporation of PDEO results in an acceleration of the first degradation step during thermogravimetric analysis in the nitrogen atmosphere. In the second step, the slower degradation rate RTmax2 (%/min) and higher char yield were observed. The mechanism of the phosphorus polyol effect during combustion was to reduce the flammability in the gas phase by cutting off oxygen access. The values of the LOI test increased with higher PDEO content in foams. LOI values were additionally tested after aging at 140 °C. Material containing 10 php of PDEO (FPUF-10) was compared with material containing 10 php of commercial additive flame retardant–dimethyl methylphosphonate (DMMP) [67,68,69]. After 16 h of aging, the FPUF-10 still exhibited significant fire resistance, and samples with commercial flame retardant had reduced fire resistance attributed to DMMP migration during aging. Along with the increase in the amount of PDEO, lower values of the fire growth rate were obtained, indicating greater fire safety of materials containing phosphorus polyol. Unfortunately, the phosphorous fragments released during the pyrolysis of PDEO restrained the ignition of combustible volatiles and resulted in more incomplete combustion; thus, the amounts of CO and CO_2_ were increased even twice for polyurethane containing 15 php of PDEO compared to the neat sample (0 php PDEO).

Chiu et al. [27] synthesized a novel flame retardant (HMCPP) containing phosphorus units and reactive hydroxyl groups and used it to obtain elastomer polyurethanes. HMCPP was achieved through the esterification of 2-carboxyethyl(phenyl)phosphoric acid and trimethylolpropane. Then, HMCPP was reacted with polycaprolactone diol (PCL) and 4,4′-diphenylmethane diisocyanate (MDI) to obtain the prepolymer. After that, 1,4-butanediol (BDO) was added as a chain extender to produce the polyurethane. The amount of phosphorus for HMCPP-containing materials ranged from 1.36 to 4.22 wt.%. As the amount of HMCPP increased, smaller amounts of chain extender were added, which resulted in polyurethane thermal stability reduction. However, the highest residue values were recorded for the samples with the highest phosphorus amounts. LOI values were in the range of 27.7% to 31.8% and increased with higher HMCPP content. All materials containing incorporated phosphorus atoms were classified as V-0 in the UL-94 rating. The HMCPP contributed to the char formation, which protected materials from flame and oxygen access.

Zhang et al. [70] obtained a novel liquid reactive polyester flame retardant (DMDP) from the reaction of dimethyl methylphosphonate with 1,6-hexanediol. Then, they prepared hot melt polyurethane adhesives containing from 0 to 6 wt.% DMDP and material containing only additive flame retardant—4 wt.% of dimethyl methylphosphonate (DMMP). Mechanical properties (tensile strength and elongation at break) and thermal stability in the N_2_ atmosphere during thermogravimetric analysis (TGA) were improved by reactive flame retardant usage compared to the material containing DMMP. However, with the increasing amount of DMDP in polyurethane adhesives, the values of 10% mass loss decreased, but char yield formation was improved. The LOI value for materials with 4 wt.% DMDP was 25.6% while for polyurethane adhesive with 4 wt.% DMMP it was 23.5%. With the increasing DMDP content, improvement in flame retardancy was observed (LOI value for 0 wt.% and 6 wt.% DMDP was 17.2% and 28.2%, respectively).

K. Gosz et al. [45] examined the impact of a mixture of phosphorus-containing polyol (Exolit^®^OP 560) and bio-polyol derived from crude glycerol or liquefied cellulose on rigid polyurethane–polyisocyanurate foams. Materials were prepared by using 0, 25, 50, 75, and 100 wt.% of a bio-polyol in the mixture with phosphorus-containing polyol. Researchers observed that Exolit^®^ OP 560 addition led to an increment in average cell diameter. Thermogravimetric analysis results showed that phosphorus-containing polyol accelerates degradation by shifting T1max and T2max to lower temperatures. However, the highest temperatures of 5 and 10% mass loss had the sample with 100 wt.% of phosphorus polyol. All the materials containing Exolit^®^ OP 560 were classified as V-0 in the UL-94 vertical test; moreover, with increasing phosphorus polyol content, the flame spread range was shorter.

F. Tabatabaee et al. [7] used the same phosphorus-containing polyol to obtain flame retardant waterborne polyurethanes. The authors additionally attached alkoxysilane groups to the WPU chains by using (3-aminopropyl)triethoxysilane to form a stable siloxane network to enhance mechanical properties. The results indicated that the sample containing 4.3 wt.% phosphorus polyol and 4 wt.% renewable castor oil polyol had the highest tensile strength equal to 37 MPa. As Exolit^®^ OP 560 content increased from 0 to 8.7 wt.%, the char residue at 600 °C increased from 2.5 to 5.37 wt.%, respectively. The LOI value of the WPU without Exolit^®^OP560 was 26.8%. However, during the UL-94 test, the drip was observed. The reason for the high LOI value was due to the siloxane crosslinking network. After phosphorus atom incorporation, flame retardancy improvement was observed—the LOI values were from 29.6–30.4% and samples were classified as V-0 after the UL-94 test. Morphology investigated by SEM showed that samples without phosphorus exhibited many tiny holes on the surface of the char caused by the intensive release of volatile gases; however, after incorporation, the char was compact and could perform as an effective barrier. The incorporation of 8.7 wt.% of Exolit^®^ OP 560 reduced the heat release rate peak (pHRR) from 302 W/g (for the neat sample) to 204 W/g. TG-IR analysis showed that during the decomposition of polyurethane containing phosphorus polyol, the phosphate groups (phosphoric acid and polyphosphoric acid) were produced. The authors concluded that flame retardancy was obtained from protective char through the synergistic effect of phosphorus and silicon and also the gas dilution effect.

L. Gu and Y. Luo [51] obtained waterborne polyurethanes modified by Exolit^®^ OP 550 and octahydro-2,7-di(N,N-dimethylamino)-1,6,3,8,2,7-di-oxadiazadiphosphecine (ODDP). Exolit^®^OP 550 was added to the prepolymer as a soft segment and ODDP was incorporated as a chain extender—a part of hard segments. The real content of phosphorus in the samples was tested by inductively coupled plasma atomic emission spectroscopy (ICP-AES) and it was in the range from 2.59% to 3.88%. During thermogravimetric analysis, a three-step degradation process was observed. The first decomposition peak was attributed to the degradation of flame retardants and with an increase in the content of Exolit^®^ OP 550, characteristic temperatures of mass loss were decreased. Phosphoric acid, which is one of the products of phosphorus polyol decomposition, can catalyze the breakdown of polyurethane. The second peak of decomposition corresponded to the depolymerization of hard segments and the last peak corresponded to soft segments. All the samples containing Exolit^®^ OP 550 were classified as V-0 after the UL-94 test. The LOI values were improved when the phosphorus polyol content increased. The highest LOI value, which equaled 31.4%, was characterized by the sample with 13 wt.% of Exolit^®^ OP 550. The cone calorimetry parameters of TTI, pHRR, MAHRE, THR, and av-EHC were reduced significantly with increasing content of phosphorus polyol. The pHRR value for the sample containing only 4.5 g of ODDP was 531.7 kW/m^2^; however, for polyurethane, which had 6.5 g of Exolit^®^ OP 550 and 4.5 g of ODDP, the pHRR value was 264.0 kW/m^2^. The synergistic effect of flame retardants was achieved. With the increase in Exolit^®^ OP 550 content, the char yield was improved and the surface of the chars became more smooth and compact. 

Y. Yuan et al. [34] obtained two new polyols, the first contained incorporated phosphorus atoms–bis(4-hydroxybutyl) phenylphosphonate (BHPP) and the second contained nitrogen atoms–melamine-derived polyol (MADP). BHPP was synthesized through a dehydrochlorination reaction of benzene phosphorus oxychloride and 1,4-butanediol. MADP was obtained in the reaction of melamine, paraformaldehyde, and diethanol amine. The authors investigated combustion behaviors and thermal stability of rigid polyurethanes containing polyether polyol LY-4110 and flame retardant polyols added separately or together. All the samples contained expandable graphite (EG) as an additive flame retardant. The synergistic effect of BHPP and MADP with EG was examined. The TGA results showed that the material with a 1:1 ratio of BHPP:MADP had the highest T50% values, both in nitrogen and air atmosphere, which equaled 405 °C and 522 °C, respectively. For this material, the char residue amounted to 39.7% under inert and 26.0% under oxidative atmospheres and LOI was the greatest (33.5%). With the increase in the content of MADP, lower pHRR and THR values were reported. MADP could promote better cross-link concentration and the formation of a protective, compact char layer. SEM images of residual chars showed that EG cannot achieve an effective char layer because the material with only EG had a lot of cracks on the surface. The material with a BHPP:MADP ratio equal to 1:1 formed a compact char layer because acids generated by BHPP could react with melamine derivative to generate salt, which covers the surface of carbon residue. Raman spectra showed that this material also had a higher graphitization degree of the char layer. Furthermore, the authors defined a potential mechanism of degradation beginning from the catalytic effect of phosphoric acid or phosphate acids, which convert degradation products into char. During that time, the crosslinking reaction between phosphorus and nitrogen-containing polyols and isocyanate leads to the formation of O=P-O- and triazine groups. Then, the NH_3_ gases released from MADP accelerate and form the char layer, and then the EG expands the layer as a consequence of the swelling process.

S. Yan and co-workers [71] synthesized reactive low-molecular flame retardant polyester diol BEOPMS by the esterification reaction of 9,10-dihydro-10-[2,3-di(hydroxycarbonyl)propyl]-10-phospha-phenanthrene-10-oxide (DDP) with diethylene glycol. Rigid polyisocyanurate-polyurethane foams (PIRs) were prepared with the use of a synthesized flame retardant via a one-step process. The effect of synthesized flame retardant was tested. The limiting oxygen index (LOI) and char residue yield value of flame-retarded rigid polyurethane foam equaled 21.0 vol.% and 21.9 wt.%, respectively, for foam with 3.5 wt.% phosphorus content compared to the reference sample (pure rigid polyurethane foam) (17.0 vol.% and 16.3 wt.%, respectively), which verified the great effect of BEOPMS on improving the flame retardancy of the foams. At a 3.5 molar ratio of NCO/OH groups and 3.5 wt.% content of phosphorus, an even higher LOI value of PIR (28.0 vol.%) with the highest char residue yield (44.7 wt.%) was obtained. Moreover, it was recognized that the isocyanurate ring groups can improve the thermal stability of prepared samples and increase the char residue yield in combination with a higher content of PMDI. 

H. Wang, Q. Liu, and X. Zhao et al. [72] also synthesized novel flame retardant polyester diol (FRPE) based on DDP, adipic acid, ethylene glycol, and 1,4-butanediol. Researchers investigated the impact of FRPE on selected properties of the flame-retarded polyurethane elastomers obtained with the use of FRPE as a polyol, 4,4-diphenylmethane diisocyanate (MDI) and 1,4-BDO as a chain extender. The selected mechanical and thermal properties and flame retardancy of the resulting FR-PUEs were characterized. It was suggested that DDP would increase the interaction between hard and soft segments, resulting in FR-PUEs of unexpected higher tensile strengths up to 40.0 MPa. Furthermore, it was found that the employment of DDP led to the residual char yield increasing the limiting oxygen index (LOI) growth and reducing the total heat release (THR). Using a low phosphorus content of 0.72 wt.%, the materials reached the vertical burning level at V-0, LOI at 24%, and THR of 52.6 MJ/m^2^. 

Table 1 presents the above-described phosphorous-containing reactive flame retardants, their common application in polyurethane materials, and selected flame test data.

## 6. Nitrogen-Containing Reactive Monomers

Nitrogen-containing compounds are a rapidly growing group of flame retardants. The main advantages of their use are that they are environmentally friendly, with low smoke evaluation, low toxicity, a solid state, and a lack of carbon dioxide, water, and halogenated acids as combustion products. The efficiency of these compounds is placed between halogenated and metallic hydroxide. They split off ammonia, hydrogen cyanide, nitrogen dioxide, and nitrogen oxide. It is astonishing that this process continues after the explosion of fire, and these compounds show a low tendency to corrode with fire hazards. Nitrogen-containing FRs can also act physically and absorb a huge amount of energy (in an endothermic reaction) by undergoing a sublimation process. For example, melamine decomposes at a high temperature and releases ammonia gas, which dilutes oxygen and other combustible gases. Moreover, melamine forms a protective carbonaceous layer in the condensation phase [73,81].

Li et al. [44] synthesized a green reactive flame retardant polyether polyol based on melamine (GPP). At first, they reacted dialdehyde starch, melamine, and distilled water, using triethanolamine as a catalyst, to obtain an environmentally friendly melamine resin. Next, they prepared GPP by reacting this melamine resin with propylene oxide and diethanolamine. The obtained polyol was used to prepare rigid polyurethane foams (RPUFs). The LOI values of foams increased linearly with a higher amount of flame retardant polyol. The sample with the highest GPP content reached an LOI value equal to 30.4%. GPP promoted the carbon generation rate, which improved the flame retardancy of foams without significantly losing their compressive strength. Moreover, GPP improved the thermal insulation performance of polyurethanes. With increasing GPP content, more carbon residue in the thermogravimetric analysis was observed.

Y. Yuan et al. [34] also obtained a melamine-derived reactive flame retardant. Nitrogen-containing FR, called MADP, was synthesized in the reaction of melamine, paraformaldehyde, and diethanol amine. The authors investigated the impact of two new polyols on the flame retardancy of rigid polyurethane foams: first, a material containing incorporated phosphorus atoms (BHPP) and, second, a material containing nitrogen atoms (MADP). The investigation results were described in the ‘Phosphorous-containing reactive monomers’ part of their study.

H. Zhu et al. [46] synthesized modified resin (EMF) using ethylene glycol, paraformaldehyde, and melamine and used it in the presence of polyether polyols, polyarylpolymethylene isocyanate, and also two additive flame retardants, ammonium polyphosphate (APP) and dimethyl methylphosphonate (DMMP), to obtain rigid polyurethane foams. Flammability test results showed that due to the presence of melamine in EMF, the LOI value was higher for PU, which contained only EMF (24.8%) compared to common LOI values for usual rigid polyurethane foams (about 19%). Materials containing APP or DMPP separately had similar LOI values. After adding APP and DMMP, the synergistic effect of flame retardants working was observed and the LOI value was the highest. Similar results were visible during cone calorimeter tests performed—the lowest pHRR and THR values were indicated for materials containing both APP and DMPP. The investigation proved that the ammonium polyphosphate promoted the formation of an expansion carbon layer because it worked in the condensed phase. During the first stage of decomposition (TGA analysis), the EMF leads to the generation of melamine, hydrogen cyanide, ammonia, HNCO, polyols, isocyanates, and some stable intermediates. At high temperatures, the released melamine could be self-condensed into melem and this structure causes the carbon layer to be more compact.

Zhu H. and co-workers [82] also investigated a series of ethylene glycol-modified urea–melamine–formaldehyde resins (EUMFs) and their effects on the cell morphology, compressive strength, flammability, combustion behavior, and thermal stability of rigid polyurethane foams (RPUFs). EUMFs were synthesized from urea, melamine, paraformaldehyde, and ethylene glycol, and then incorporated into RPUFs as a reactive-type liquid flame retardant. The results indicate that the incorporation of EUMFs can adversely affect cell morphology and compressive strength of RPUFs; nevertheless, it results in an increase in the LOI value of RPUFs to about 24%. The cone calorimetry test results reveal that RPUFs exhibit good fire resistance and smoke suppression. 

Zhang J., Zhou Y., and co-workers [83] investigated a flame retardant polyol with a low viscosity prepared from cardanol and melamine–cardanol-derived Mannich base polyol (MCMP) and RPUFs prepared using this FR polyol. In addition to using the flame retardant polyol for RPUFs, the synergistic effects of adding an ammonium polyphosphate (APP) and diethyl ethylphosphate (DEEP) and commercial unmodified EG to the foams were also investigated. Mechanical properties, thermal stability, and fire properties of the new RPUFs were studied in detail. The results indicate that the introduction of a melamine moiety to the molecular structure of MCMP enhanced the mechanical properties, thermal stability, and flame resistance of the resulting rigid polyurethane foam (RPUF). The thermal conductivity of pure RPUF equaled 0.022 W/m-K, which was a lower value than that of FR-filled RPUFs. The authors affirmed that thermal conductivity increased with FR content due to the fact that the solid FRs damaged the structure of cells. The reference sample (pure RPUF) exhibited an LOI of 22.5%. With the addition of only 10 php of FR in the formulations, the LOI increased to 24.5%, 25.2%, and 26.4% for the DEEP, APP, or EG-filled RPUFs, respectively. It is also noted that the LOI increased continuously with the increase in FR content. Furthermore, compared to neat RPUFs, the FR-filled RPUFs exhibited increased compressive strength, thermal stability, and char residue and also reduced heat release and smoke emission during combustion. The improved flame retardancy is most likely related to the formation of a reinforced char layer after combustion.

Table 2 presents the above-described nitrogen-containing reactive flame retardants, their common application in polyurethane materials, and selected flame test data.

## 7. Reactive Mono-Component Flame Retardant Monomers Containing Multiple Flame Retardant Groups

Recently, mono-component flame retardants containing multiple flame retardant groups, for example, phosphorus–nitrogen (P/N), phosphorus–silicon (P/Si), silica–nitrogen (Si/N), or phosphorus–nitrogen–silica (P/N/Si), have become popular due to their high efficiency caused by a synergistic effect of different flame retardance elements and structures [84]. Haojun Fan and co-workers [85] synthesized a novel flame retardant with dihydroxyl groups via reaction of the DOPO with Schiff base that was utilized as a copolymeric component for the preparation of flame-retarded waterborne polyurethanes, FRWPUs. Simultaneously, to balance the cold resistance and mechanical properties of the WPU and consider the synergy of P and Si, HO-Si was conjugated into the main chain of WPU. Compared with pure WPU, the LOI value of FR/Si-WPU increased and the UL 94 rating was upgraded to V-0. It was found that the PO⋅ radicals generated during decomposition can capture H⋅ and OH⋅ radicals, which then decrease the flame propagation rate, and the Si can produce a special insulative Si–O layer to improve the thermo-oxidative stability. Moreover, the synergistic effect of P and Si can form denser and graphitized chars that improve flame retardancy.

Yinghong Chen et al. [86] designed and successfully synthesized novel multi-structure synergistic graphene-based ternary graft flame retardant oapPOSS-graft-GO-graft-DOPO(PeGeD). Researchers investigated the synergistic effects of graphene/nitrogen/phosphorus/silicon multi-structures on reducing fire hazards. It was described that the graphene sheets enhance the barrier property of carbon layers to prevent heat and volatile transfer; the phosphorus-containing groups as catalysts accelerate and promote the char formation in the condensed phase; the SieOeSi structures improve the thermal oxidative stability; and the nitrogen elements in PeGeD produce the inert gases at high temperature and play a major role in the gaseous phase.

Haojun Fan et al. [87] also investigated the synergistic effect of phosphorus–nitrogen and silicon-containing chain extenders on the mechanical property, flame retardancy, and thermal degradation behavior of waterborne polyurethane. Thermal degradation behavior and residue char analysis demonstrate that the excellent flame retardancy of the prepared FRWPU is attributed to the synergistic effect of phosphorus–nitrogen and silicon-containing reactive flame retardants in the condensed phase, which produce a more compact, intumescent, and thermal stable graphitic structured char with –P(=O)–O–Si– linkages, acting as a “tortuosity effect” to delay the escape of volatile degradation products and heat transfer. The described approach offers a promising strategy for addressing the current bottleneck problem of most existing phosphorous-based flame retardants that enhance flame retardancy but at the expense of the thermal and mechanical properties of polyurethanes.

Obtaining materials using reactive mono-component flame retardant monomers containing multiple flame retardant groups is a huge challenge. The influence of the chemical structure of monomers on the selected functional properties of materials is significant, which is why it is important to develop monomers that will not change the properties of materials. Hence, most scientific works can be found describing the use of mono-component flame retardants containing phosphorus–nitrogen (P/N) retardant groups.

Phosphorus–nitrogen intumescent flame retardants (IFRs) are characterized by low corrosion, toxicity, smoke production, and also high fire resistance. Unfortunately, additive IFRs have weak interaction and low affinity with organic substances of polyurethane. P. Zhang et al. [49] obtained a reactive IFR named BSPB, which was synthesized from phosphorus oxychloride, pentaerythritol, and benzoguanamine. BSPB was used as a chain extender in waterborne polyurethanes and added in an amount ranging from 0 to 8 wt.%. A material containing 8% BSPB was classified as V-0 during the UL-94 vertical burning test and had an LOI value equal to 27,3%. The other samples containing 0, 2, or 4 wt.% of BSBP dripped and ignited the cotton during the burning test. The TGA results showed that PU films with BSPB contributed little negative effect on initial decomposition temperatures compared to neat PU, but they exhibited higher temperatures of decomposition peaks and more char residues were noted. The incorporation of BSPB into the polymer chain resulted in improvements in mechanical properties and tensile strengths for materials with 0 wt.% and 8 wt.% of reactive chain extender, with values of 15.3 MPa and 26.4 MPa, respectively. This phenomenon can be correlated with the presence of additional amide linkages formed by amino and isocyanate groups and with the conjugation of the heterocyclic triazine ring structure in BSPB.

Rigid polyurethane foams are promising insulation materials due to their low thermal conductivity. To make these materials useful in building constructions, they cannot be flammable. R. Yang et al. [84] synthesized hexa-(5,5–dimethyl–1,3,2–dioxaphosphinane–hydroxyl–methyl-phenoxyl) cyclotriphosphazene (HDPCP) and added it as a reactive flame retardant to obtain rigid polyurethane foams. Samples containing 0 wt.% and 15 wt.% of HDPCP had thermal conductivity values equal to 0.0275 W/m^2^K and 0.0268 W/m^2^K, respectively. LOI values increased with a higher amount of flame retardant and ranged from 19% to 25%. TGA data showed a trend of improvement in T5% and char yield with growing HDPCP content. Cone calorimeter test results showed that the pHRR value for reference PU was 235.7 kW/m^2^ and for material with 15 wt.% of HDPCP it was equal to 154.5 kW/m^2^. Moreover, the reduction in total smoke release (TSR) and total smoke production (TSP) parameters for this HDPCP-containing polyurethane was registered.

S. Wang et al. [50] obtained two IFR flame retardants, pentaerythritol di-N-hydroxyethyl phosphamide (PDNP) and a novel monomer tri(N, N-bis-(2-hydroxy-ethyl) acyloxoethyl) phosphate (TNAP). PDNP was obtained in a multistep reaction with, among others, pentaerythritol, trichlorophosphate, and ethanolamine, but TNAP was synthesized by a reaction of phosphorus oxychloride, hydroxyethyl acrylate, and diethanolamine. They have since used these reactive substances as substrates for flame-retarded waterborne polyurethane syntheses. The highest LOI value (25.5%) was attributed to materials with 4 wt.% TNAP and 3 wt.% PDNP. During thermogravimetric analysis, hard segments were decomposed between 270 °C and 350 °C and the second step of degradation was registered from 350 °C to 450 °C for all the materials. TNAP contributed to the reduced thermal stability of PU in the initial stage of decomposition, which was attributed to the low degradation temperature of the O=P-O bond occurring in this flame retardant. SEM micrographs after LOI tests showed that the sample without flame retardants had a cracked and neat surface, which was incapable of protecting the PU matrix. The sample containing 2 wt.% TNAP and 3 wt.% PDNP had a wrinkled surface with interlaced honeycomb pores covering the char. The authors concluded that TNAP works synergistically with PDNP and these FRs, forming a protective layer against sudden volatilization pyrolysis products from entering the flame zone. The highest tensile strength had material containing only 3 wt.% of TNAP; this result was probably due to the branched structure of this FR, which provided a highly crosslinked polyurethane structure, while the PDNP showed a plasticizing effect.

H. Ding et al. [88] synthesized bio-based reactive flame retarding polyol (FRPE) by firstly obtaining castor oil-based fatty acid amide and next reacting it with 9,10-dihydro-9-oxa-10-phosphaphenanthrene-10-oxide and triphenylphosphine. They also acquired another reactive flame retardant, dihydric alcohol (BHAPE), from diethanolamine, formaldehyde, and diethyl phosphite. The optimal BHAPE to FRPE ratio was found to be 4:6 because when the amount of BHAPE in PU foam exceeded 40 wt.%, the compressive strength decreased quickly and the LOI value was high (equal to 23%). Above 60 wt.% of FRPE, there were issues with polyurethane mixing due to the high viscosity of the system. The authors compared polyurethanes obtained with a commonly known polyether polyol named 4110 (4110/PUF) with PU foam based on 4 wt.% of BHAPE and 6 wt.% of 4110 (4110/BHAPE/PUF) and material containing 4 wt.% of BHAPE and 6 wt.% of FRPE (FRPE/BHAPE/PUF). After incorporating BHAPE into the polymer chain, compared to the neat PU, pHRR values were reduced by 26.7% and 61.1% for 4110/BHAPE/PUF and FRPE/BHAPE/PUF, respectively. Combustion behaviors showed that FRPE and BHAPE accelerated CO production but reduced CO_2_ emission due to incomplete combustion of flame-retarded polyurethanes. Both BHAPE and FRPE enhanced the thermal behaviors of PUs at temperatures from 400 °C to 800 °C, which was confirmed by the TGA. These FRs promote char formation at low temperatures, which could protect materials at higher temperatures. TGA-FTIR and TGA-MS results showed that FRPE and BHAPE inhibited the release of pyrolysis products at low temperatures. +PO, HOP, +PO_2_, PO_2_H_3_, and PO_3_H_3_ fragments derived from FRPE and BHAPE degradation were registered. Moreover, the authors proposed a probable FRPE/BHAPE/PUF degradation mechanism: first P-O-C, P-N, and P-C bonds were broken in polyurethane and next the PO•, P•, H•, and HO• free radicals were formed. With the temperature increasing, decomposition of aliphatic and aromatic compounds was registered and polyphosphoric acid could be formed, which accelerated the char formation. In the end, water, carbon monoxide, and carbon dioxide were volatilized, and a stable char layer was created. Z. Lu et al. [52] also synthesized BHAPE from diethanolamine, formaldehyde, and diethyl phosphite and added it to NCO-terminated prepolymers to obtain flame-retarded reactive hot melt polyurethane adhesives (HMPURs).

A. M. Borreguero et al. [47] received aminophosphonate polyether polyols (PFyX, where X is the cation of catalyst) in a reaction using diethyl bis(2-hydroxyethyl) aminomethyl phosphonate with different catalysts: CsOH, KOH, NaOH, MeOK, distillate water, and propylene oxide. The obtained polyols were used in conjunction with commercial polyol Alcupol R-4520 to synthesize polyurethane foams. FTIR analysis confirmed phosphorus polyol incorporation. PFyCs showed a better polymerization rate compared to the other polyols synthesized with H_2_O and metal hydroxides. PU with the content of PFyCs from 0 pph to 25 pph had 7.79–22.13% of char residue at 600 °C during thermogravimetric analysis, respectively. Comparing two samples with 0 and 25 pph of PFyCs, cone calorimeter results showed that pHRR, THR, TSP, and TSR were reduced about twice for material containing 25 pph of aminophosphonate polyether polyol.

Luo Y. et al. [89] synthesized a phosphoramide monomer with a ringed structure by the reaction of phosphoryl chloride and dimethylamine hydrochloride with ethanoloamine in dichloromethane solution. P–N synergistic halogen-free flame retardant WPU (DPWPU) was prepared by reacting the ODDP with toluene diisocyanate TDI, polyoxypropylene glycol PPG1000, and dimethylol propionic acid DMPA. Limiting oxygen index (LOI) tests confirmed increasing values with higher ODDP content at DPWPU up to 15 wt.%. For the sample with 15 wt.% of ODDP content, the LOI of DPWPU equaled 30.6%, whereas for 20 wt.% of ODDP, the LOI decreased to 29.8%. Materials with 10, 15, and 20 wt.% of ODDP content characterized V-0 classification at the UL-94 test. Thermogravimetric analysis and scanning electron microscopy suggest the excellent flame retardancy of the DPWPU polymer. Compared with the unmodified WPU, the thermo-decomposition temperature of the DPWPU was reduced and the amount of carbon residue was increased to 18.18%. The surface of carbon residue was shown to be compact and smooth without holes, which would be favorable for resisting oxygen and heat.

Yang A-H. and co-workers [90] investigated Schiff-base polyphosphate ester (SPE) flame retardant synthesis through melt polycondensation. The obtained SPE flame retardant was used to prepare flame-retarded thermoplastic polyurethane elastomer (TPU). In UL-94 tests, the flame-retarded TPU containing 5 wt.% of SPE was classified as V-0 rating with a limited oxygen index (LOI) of 29%. In the cone calorimetry test, the peak of heat release rate (pHRR), average mass loss rate (AMLR), and maximum average rate of heat emission (MARHE) of the flame-retarded TPU with 5 wt.% SPE was respectively decreased by 61.7%, 41.8%, and 30.2% compared with the corresponding value of neat TPU. Moreover, THR values were also reduced for TPU with 5 wt.% of SPE compared to pure TPU, from 83.1 MJ/m^2^ to 69.1 MJ/m^2^, respectively. All of these results demonstrated that the SPE simultaneously improved the flame retardancy and dripping behavior of TPU. 

Table 3 presents the above-described reactive mono-component flame retardant monomers containing multiple flame retardant groups, their common application in polyurethane materials, and selected flame test data.

## 8. Silicon-Containing Reactive Monomers

Silicon-containing compounds could improve the thermal stability and flame resistance of polyurethanes. The benefits of such compounds are attributed to the decrease in flammable organic components and also the propagation of the isolating silica layer on the surface of the material, which inhibits heat flow. Furthermore, Si-containing components are environmentally friendly and combustion of these materials is not accompanied by corrosive smoke [95].

A commonly known method of using silicon atoms to reduce the flammability of polyurethane materials is the incorporation of these atoms by inorganic fillers. Talc, mica, or montmorillonite have hydrophilic properties. These fillers require additional intercalation with molecules of amino acids, ammonium, or phosphonium salts to make them compatible with polyurethanes. Otherwise, phase segregation can occur, which reduces the performance properties of the composite [48].

X. Hong et al. [96] synthesized reactive silicate-reinforced polyurethane composite during the reaction of toluene 2,4-diisocyanate, polypropylene glycol, and hydroxyl orthosilicic acid, which brings the Si-O bond into the main chain of PU. The research showed a significant improvement in the mechanical properties and flame resistance. The tensile strength of the material containing 20 wt.% silicate was improved to 168% compared to pure PU. Materials with silicate FR content higher than 20 wt.% were classified as V-0 in the UL-94 test, with LOI values of 32.6–34.5%. Moreover, TG curves showed that the thermal stability of the PU/silicate composite deteriorated and the residue rate increased significantly compared to pure PU.

The research and development of new polyol raw materials have led to the use of metasilicic acid (MSA) to obtain polyetherols containing silicon heteroatoms. J. Paciorek-Sadowska et al. [48] obtained polyether polyols based on metasilicic acid (MSA). MSA-based polyol was synthesized by using metasilicic acid, glycidol, and ethylene carbonate. The content of Si atoms in MSA-based polyol equaled 2.32%. Moreover, the obtained polyol together with the product of poly(lactic acid) waste glycolysis was used as a polyol mixture for the polyurethane foam preparation. PU foams based on polyol mixture showed better thermal resistance in the TGA measurements and longer time to ignition, but lower LOI and higher HRR and THR values in comparison with the foam based on MSA-polyol.

A new and promising group of materials are polyurethanes, including polyhedral oligomeric silsesquioxanes (POSSs), which have a possibility of chemical bonding with an organic PU matrix when the POSS contains reactive hydroxyl, amino, or isocyanate groups. These organic substituents enable the synthesis of organic–inorganic polyurethane hybrid materials. POSS molecules can be incorporated as side groups in the polyurethane chain (monofunctional POSS), in the main chain fragment (di-functional POSS), or as cross-linking agents (multifunctional POSS) [35]. Silsesquioxanes (POSSs) can be used as thermal resistance modifiers of polyurethanes due to the presence of hydroxyl groups at the end of chains. Hydroxyl-terminated macromolecules can react with isocyanate groups of the urethane to form prepolymers leading to polyurethanes with enhanced thermal resistance. The main chain of siloxane is the Si-O bond, which ensures chain flexibility. Moreover, the high bond energy results in significant thermal stability of polyurethane under high temperatures. The structure of siloxane is also unique in terms of hydrophobicity, weather resistance, high gas permeability, low toxicity, insulating properties, and excellent resistance to ultraviolet radiation [33,36].

M. Szołyga et al. [33] investigated the effect of octakis[(3-hydroxypropyl)dimethylsiloxy] octasilsesquioxane (SF) in addition to the selected properties of the PU materials. SF was added in an amount of 1, 3, or 5 wt.% to the PU based on hexamethylene diisocyanate (HDI) and 1,6-hexanediol (HDO). It has been proven that the use of 3 wt.% of SF contributed to the best thermal properties. The TGA test showed an increase in each of the T5% and T10% temperatures by approximately 15% compared to the reference sample (without POSS addition).

Liu et al. [36] synthesized cast polyurethane materials based on functionalized polydimethylsiloxane (FPDMS), isophorone diisocyanate (IPDI), dibutyltin dilaurate (DBTDL) as a catalyst, and 1,4-butanediol (BDO) as a chain extender. The same authors also reacted 3-aminopropyltriethoxysilane (APTS) with 3-glycidoxypropyltrimethoxysilane (GPTS) to produce bridged polysilsesquioxanes (APTS-GPTS) and added them to the previously mentioned prepolymer and extended the material with BDO. The synthesized materials were named Si-PU and Si-PU/APTS-GPTS, respectively. The TGA results demonstrated improvements in 5% weight loss temperature, maximum degradation rates, and char yield formation for Si-containing samples, with the best findings observed for Si-PU/APTS-GPTS materials. The benzene ring contributed to the char yield formation to prevent oxidation of the sample. The compact SiO_2_ structure protected the internal part of the polyurethane.

J. Pagacz et al. [40] obtained PU elastomers containing 1,2-propanediolizobutyl POSS (PDI-POSS), disilanollsobutyl POSS (DSI-POSS), or (octa(3-hydroxy-3-methylbutyldimethylsiloxy) POSS (OCTA-POSS). Their chemical incorporation into polyurethane structure functionalized oligomeric silsesquioxanes as a pendant group (PDI-POSS), part of the main chain (DSI-POSS), or as chemical crosslinkers (OCTA-POSS). The TGA results in the inert atmosphere showed that PDI-POSS polyurethane had higher temperatures in the initial stage of degradation compared to DSI-POSS and OCTA-POSS elastomers. However, the second one stabilized the final decomposition of materials and showed higher amounts of char residues.

In the last 5 years, there has been a lack of articles reporting results on the flame resistance of polyurethanes containing only Si atoms from embedded POSSs in their structure. In 2009, S. Bourbigot et al. [97] investigated the reaction to fire of thermoplastic polyurethane (TPU) containing POSSs. The incorporation of 10% of silsesquioxanes in TPU caused a decrease of 80% in the pHRR value compared to neat polyurethane, but time to ignition for TPU-POSS was two times longer and there was no significant improvement in LOI value and UL-94 results (23 and V-2, respectively). Limiting HRR values were caused by the protection of the char layer, which forms a thermal barrier limiting heat and mass transfer.

Table 4 presents the above-described silicon-containing reactive flame retardants, their common application in polyurethane materials, and selected flame test data.

## 9. Perspectives

Nitrogen, phosphorus, and silicon-containing semi-products are the most investigated and promising flame retardants for polyurethane materials. The expected idea is to incorporate different types of chemical reactants into the structure of polyurethanes to create a fire-resistant material that will be more eco-friendly than commercially used materials. In the case of reactive flame retardants, their high effectiveness with a relatively low amount of addition to the polyurethane system can be observed. The addition of reactive flame retardants is an advantage compared with additive antipyrines. Moreover, a big advantage of using reactive flame retardants is the small amount required during polyurethane preparation, which does not drastically increase the cost of producing such material.

Nevertheless, it could be desirable to establish a new type of flame retardant agent that could be obtained from different nitrogen and phosphorus-containing resources derived from substances of natural origin, the waste stream, or by-products of processing, and characterized by the same efficiency as those products currently being used. New pro-environmental and pro-economical actions are expected in the near future, especially due to the unstable world economic situation regarding fossil sources—oil and gas.

The second most advantageous solution would be to find a way to increase the flame resistance of polymer materials, thereby reducing the emission of toxic gases and carbon oxides into the atmosphere during combustion. An example of an investigation focused on bio-based reactive flame retardants is the research developed by Professor Paciorek-Sadowska’s research team [98,99,100]. Researchers synthesized a new type of bio-polyol based on unrefined white mustard (Sinapis alba) oil as a result of a two-step synthesis consisting of epoxidation of double bonds and subsequent opening of the created oxirane rings. The obtained bio-based polyols were used for rigid polyurethane foam preparation. Selected measurement results indicated a great impact of the prepared polyols on thermal properties, including fire resistance. Modified foam had better functional properties than reference foam, e.g., lower brittleness, better thermal insulation properties, and better fire resistance. Moreover, foams modified by bio-polyol based on mustard seed oil showed lower apparent density, brittleness, compressive strength, absorbability, and water absorption, as well as thermal conductivity, compared to the reference (unmodified) foams. Furthermore, the obtained materials were more resistant to aging and more susceptible to biodegradation in the soil environment.

Due to the gas-dominant mechanism of organic flame retardants, the resulting flame retardant polyurethanes always exhibit worsening smoke and toxic gas release. The hazards of toxic smoke in fires have increased significantly. Currently, methods are being sought not only to reduce the flammability of materials but also to reduce the amount of smoke emitted during the combustion of flame retardant materials. Relatively good methods for smoke minimization are surface treatment methods. Yu-Zhong Wang [101] and co-authors demonstrate an environmentally friendly and facile flame retardant nanocoating that can effectively suppress smoke and toxic gas release in fire for flexible polyurethane foam (FPUF). Researchers investigated a water-based functional coating system in which nanoporous sepiolite nanofibers were wrapped via phytic aid to have a negative charge, and then together with the positively charged CH, the nanofibers were stably deposited onto the surface of FPUF through Layer-by-Layer assembly. The smoke density and the peak production of toxic CO and NOx were greatly decreased by as much as 77%. Furthermore, the nanocoating had little effect on the mechanical properties.

On the other hand, polymer material manufacturers use halogenated organic compounds to enhance flame retardancy. Although halogenated flame retardants (FRs) are highly effective in suppressing flammability, their adverse environmental and health effects led to legislative restrictions on their utilization. Nowadays, halogen-free high flame retardance of polyurethanes without deteriorating their intrinsic performances is still a great challenge. Gexin You et al. [102] proposed a green surface flame retardant (SFR) for rigid polyurethane (RPU) foams. In detail, a new halogen-free UV-curable self-extinguished coating was developed and then combined onto the surface of RPU, obtaining a flame retardant RPU foam surface system (SFR-RPU). The results confirmed that SFR-RPU foams exhibit highly efficient flame retardancy at a coating thickness of 25 µm, showing a fast self-extinguishing behavior. A compact porous char layer serves as a good barrier that can be rapidly formed on the surface of SFR-RPU. Furthermore, when compared to neat RPU, the compressive strength of SFR-RPU foams is enhanced and the heat-insulation property is maintained perfectly. This facile, low-cost, and environmentally friendly post-treatment of RPU foam could expand its fire-safe commercial applications.

Due to the hazards associated with polyurethanes, there is a growing effort to produce them without isocyanates. These so-called non-isocyanate polyurethanes (NIPUs) rely on alternate reactions to produce carbamate moieties with different constituents. Jaime C. Grunlan and co-workers [103] investigated and described the chitosan (a natural polysaccharide) and tannic acid (a natural polyphenol) incorporated into a NIPU. These two bio-sourced ingredients have been shown to facilitate charring and improve flame resistance in other systems including PU foam. The resulting foams self-extinguish immediately following a 10 s exposure to a butane torch flame. In addition, the principal weight loss or heat release event in TGA or MCC is delayed by >100 °C compared to traditional rigid polyurethane foam. This NIPU requires no further treatments or additives and offers superior thermal performance to commercial rigid foam. Variations in this material could potentially be utilized as insulation in construction.

## 10. Conclusions

The flammability of polyurethanes strongly depends on the structure of the polyol and the isocyanate incorporated into the structure of polyurethanes via appropriate reaction. Up to now, the main monomers that are considered fire-resistant include the nitrogen atoms. As a consequence, polyurethanes contain a significant amount of nitrogen in their structures, which can lead to the production of HCN during pyrolysis or combustion. This situation can indicate some problems with the usability of some final products. On the other hand, phosphorus- or silicon-containing monomers are mostly discussed in the world literature because they are effective at relatively low content levels, and usually provide barely acceptable mechanical properties. Significant reduction in various properties of such polyurethanes can certainly result in less interest in such substances compared to those with nitrogen atoms.

Some of the commercial fire retardants contain phosphorus and halogen in the same molecule. The role of each element and the possibility of their synergism is disputed in the literature. A number of new phosphorus- or phosphorus–nitrogen-containing additives or reactive polyols are suggested in the literature for rigid PU foams. It is clear from this review that the additive flame retardant solution should be meticulously selected, not only in terms of flame retardancy efficiency but also considering other properties that should be monitored and optimized, including environmental impact. It was shown in this review that different types of flame retardant additives, especially earlier used phosphorus-containing chemicals, lead to a decrease in the thermal stability of PUs but promote cross-linking and charring. There is no single reactive flame retardant that would perform best in the case of various kinds of polyurethanes. Therefore, each polyurethane material must be analyzed separately. 

Today it is clear that incorporating appropriate flame retardant molecules as main monomers (polyols or diisocyanates) to the polyurethane is the best solution to obtain novel materials with reduced flammability with better safety. Sometimes, using different multi-atom reactants can be considered to achieve a better fire resistance index. Nevertheless, it is always important to analyze the environmental impact of gases released during combustion. This review has demonstrated that the complexity of the combustion process for fire-resistant polyurethanes modified by commercially used fire retardants makes it necessary to develop new kinds of flame retardants that are more eco-friendly. 

The latest literature reports show that eco-flame retardants for polyurethane materials at various levels of impact, reactive monomers of bio-origin, processing methods that reduce flammability, and obtaining polyurethane materials with reduced flammability without the use of toxic isocyanates are being sought. From an ecological point of view, the most advantageous approach seems to be the search for monomers of plant origin, which, by being incorporated into the structure of polyurethanes, will improve the flame resistance of the materials while maintaining their strength properties.

## Figures and Tables

**Figure 1 ijms-25-05512-f001:**
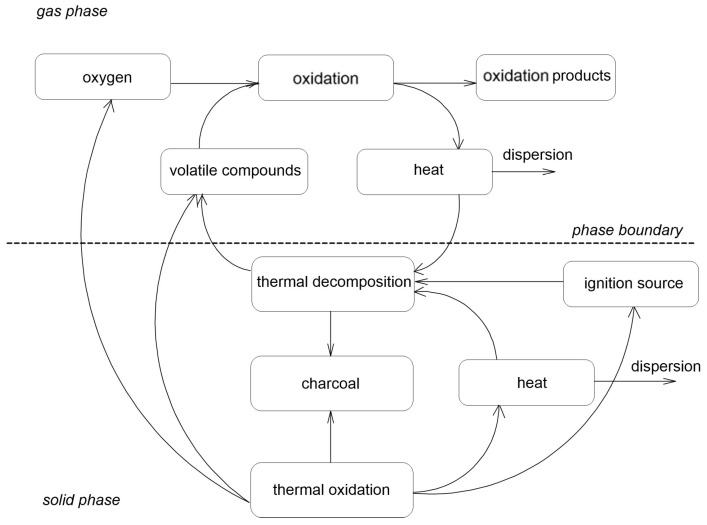
The scheme of the combustion process [13].

**Figure 2 ijms-25-05512-f002:**
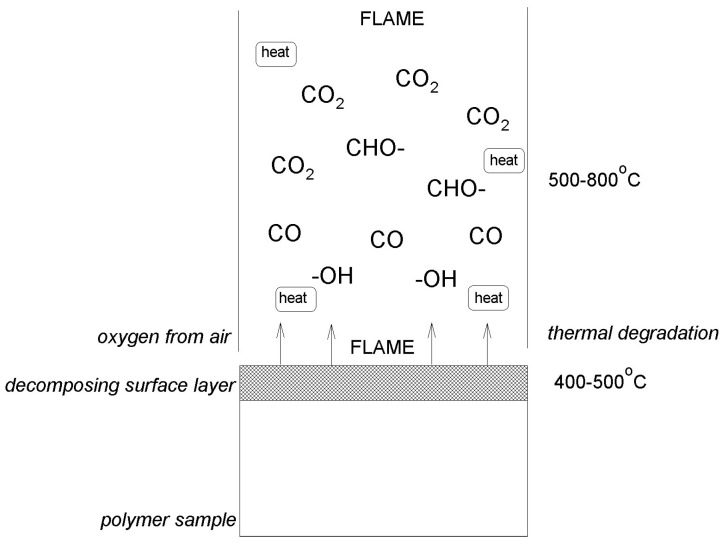
The scheme of chemical conversion during polymer combustion [14].

**Table 1 ijms-25-05512-t001:** The collection of phosphorous-containing reactive flame retardants.

No.	Type of PU	Reactive Flame Retardants (FRs) and/or Thermal Resistance Modifiers	FR Name	LOI [%]	UL-94	Char Residue Yield [%]	Total Heat Release (THR) [MJ/m^2^]	Smoking Reduction	Data Ref.	Other Refs.
**1**	Rigid foam	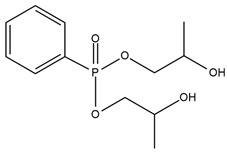	PPA-PO	No data	No data	3.3–15.5 (0–1.5 FR wt.%)	15.9–12.2 (0–1.5 FR wt.%)	No data	[24]	[24,73]
**2**	Flexible foam	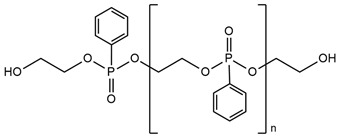	PDEO	21.7–23.0 (0–10 FR wt.%)	No data	4.3–10.1 (0–10 FR wt.%)	22.5–23.3 (0–10 FR wt.%)	Amounts of CO and CO_2_ were increased	[25]	[25,74,75]
**3**	Elastomer	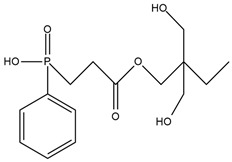	HMCPP	21.0–31.8 (0–10 FR wt.%)	V-0	No data	No data	No data	[27]	[27,76,77]
**4**	Adhesive	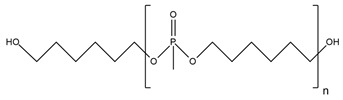	DMDP	20.8–28.2 (0–5 FR wt.%)	No data	No data	No data	Amounts CO_2_ were increased	[70]	[70]
**5**	Rigid foam, coating	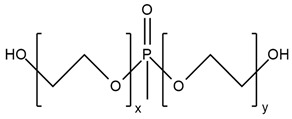	Exolit^®^ OP 560	29.6–30.4 (0–8 FR wt.%)	V-0	3.3–5.5 (0–8 FR wt.%)	27.1–25.6 (0–8 FR wt.%)	No data	[7]	[7,45]
**6**	Coating, PU resin	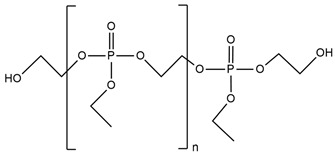	Exolit^®^ OP 550	24.5–33.0 (0–20 FR wt.%)	V-0	No data	No data	No data	[78]	[6,51]
**7**	Rigid foam	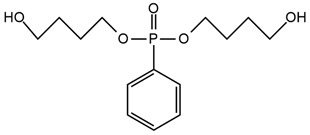	BHPP	20.0–30.3 (0–24 FR wt.%)	V-0	19.5–29.3 (0–24 FR wt.%)	47.7–29.3 (0–24 FR wt.%)	No data	[34]	[34,77]
**8**	Rigid foam, elastomers, TPU	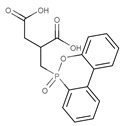	DDP	17.0–21.0 (0–28 FR wt.%)	V-0	17.8–20.6 (0–28 FR wt.%)	13.1–11.0 (0–28 FR wt.%)	No data	[71]	[71,72,77,79,80]

**Table 2 ijms-25-05512-t002:** The collection of nitrogen-containing reactive flame retardants.

No.	Type of PU	Reactive Flame Retardants and/or Thermal Resistance Modifiers	FR Name	LOI [%]	UL-94	Char Residue Yield [%]	Total Heat Release (THR) [MJ/m^2^]	Data Ref.
**1**	Rigid foam	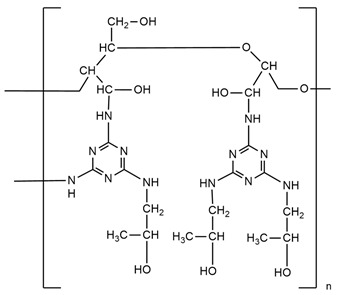	GPP	17.0–30.4 (0–48 FR wt.%)	No data	No data	No data	[44]
**2**	Rigid foam	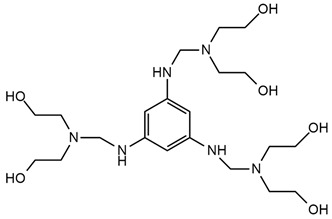	MADP	20.0–30.0 (0–24 FR wt.%)	V-0	19.5–38.4 (0–24 FR wt.%)	47.7–18.2 (0–24 FR wt.%)	[34]
**3**	Rigid foam	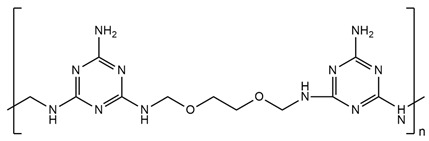	EMF	24.8 (30 FR wt.%)	No data	No data	28.1 (30 FR wt.%)	[46]
**4**	Rigid foam	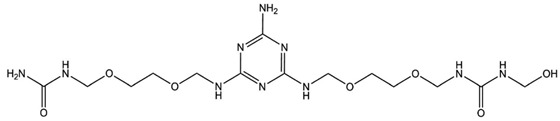	EUMF	18.1–24.4 (0–22 FR wt.%)	No data	5.4–29.8 (0–22 FR wt.%)	25.5–18.7 (0–22 FR wt.%)	[82]
**5**	Rigid foam	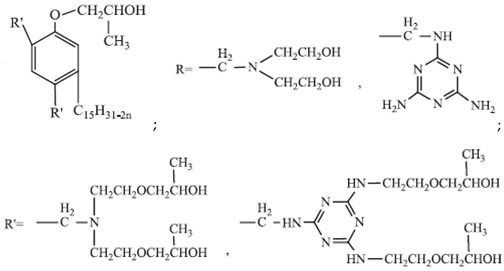	MCMP	22.5–26.0 (0–8 FR wt.%)	No data	18.8–24.4 (0–8 FR wt.%)	20.1–15.6 (0–8 FR wt.%)	[83]

**Table 3 ijms-25-05512-t003:** The collection of reactive mono-component flame retardant monomers containing multiple flame retardant groups.

No.	Type of PU	Reactive Flame Retardants and/or Thermal Resistance Modifiers	Name of FR or Modifier	LOI [%]	UL-94	Char Residue Yield [%]	Total Heat Release (THR) [MJ/m^2^]	Ref.
**1**	Coating	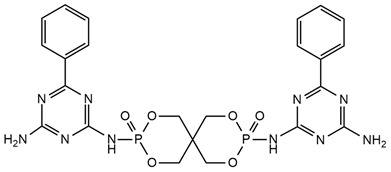	BSPB	18.6–27.3(0–8 FR wt.%)	V-0	0.7–4.7(0–8 FR wt.%)	No data	[49]
**2**	Rigid foam	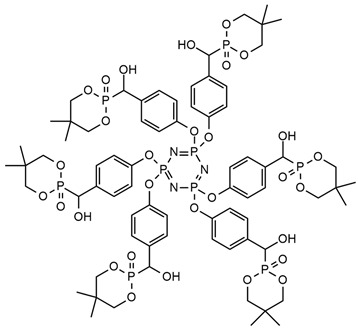	HDPCP	19.0–25.0(0–25 FR wt.%)	No data	15.8–28.7(0–25 FR wt.%)	14.7–13.6(0–15 FR wt.%)	[84]
**3**	Coating	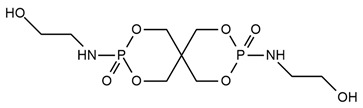	PDNP	19.5–26.0(0–9 FR wt.%)	No data	0–1.6(0–9 FR wt.%)	61–46(0–9 FR wt.%)	[50,91]
**4**	Coating	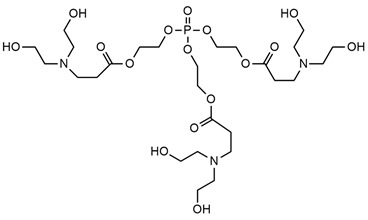	TNAP	19.5–21.0(0–2 FR wt.%)	No data	0.5–1.15(0–2 FR wt.%)	No data	[50]
**5**	Foam	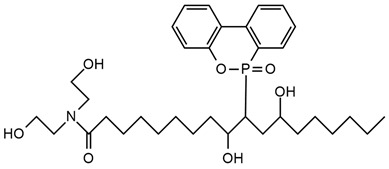	FRPE	No data	No data	No data	No data	[88]
**6**	Foam/adhesive	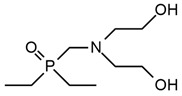	BHAPE	19–23.0(0–15 FR wt.%)	No data	No data	21.2–12.4(0–15 FR wt.%)	[52,88]
**7**	Foam	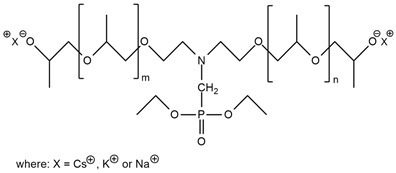	PFyX	No data	No data	No data	1.8–0.8(0–25 FR wt.%)	[47]
**8**	Waterborne PU, coatings, adhesives	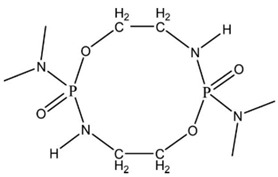	ODDP	25.4–29.8(0–20 FR wt.%)	V-0	3.0–18.2(0–20 FR wt.%)	No data	[51,89,92,93,94]
**9**	TPU	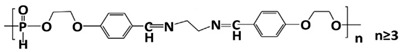	SPE	23.0–29.0(0–5 FR wt.%)	V-0	10.12–19.33(0–5 FR wt.%)	83.1–69.1(0–5 FR wt.%)	[90]

**Table 4 ijms-25-05512-t004:** The collection of silicon-containing reactive monomers.

No.	Type of PU	Reactive Flame Retardants and/or Thermal Resistance Modifiers	FR Name	Thermal Resistance Improvement	LOI [%]	UL-94	Char Residue Yield [%]	Total Heat Release (THR) [MJ/m^2^]	Ref.
**1**	Foam	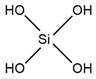	Orthosilicic acid	Accelerates the initial stage of degradation, but stabilizes the final decomposition	19.6–34.5(0–30 FR wt.%)	V-0	No data	No data	[96]
**2**	Bio-basedfoam	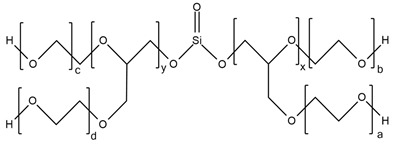	MSA-based polyol	Better for MSA-PLA polyol mixture	21.6–36.0	No data	0.5–49.6	4.7–9.7	[48,95]
**3**	Resin	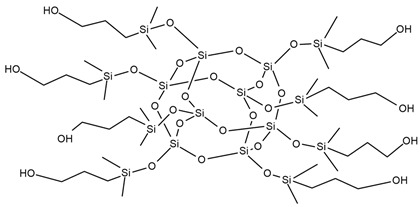	SF	Yes, with 3% of POSS	No data	No data	No data	No data	[33]
**4**	Resin	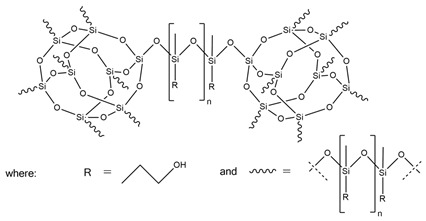	SiHQ	Yes	No data	No data	No data	No data	[33]
**5**	Cast polyurethane	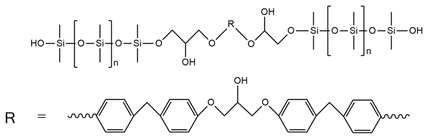	FPDMS	Yes	No data	No data	No data	No data	[36]
**6**	Cast polyurethane	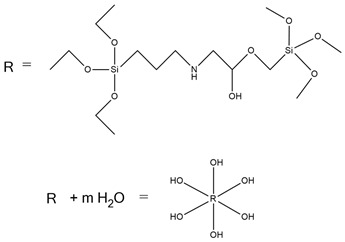	APTS-GPTS	Yes	No data	No data	No data	No data	[36]
**7**	Elastomer	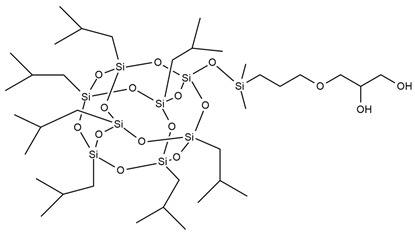	PDI-POSS	Better in the initial stage of degradation	No data	No data	No data	No data	[40]
**8**	Elastomer	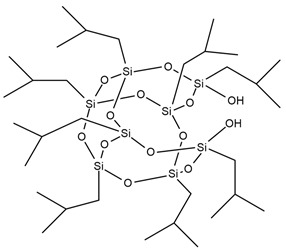	DSI-POSS	Accelerates the initial stage of degradation, but stabilizes the final decomposition	No data	No data	No data	No data	[40]
**9**	Elastomer	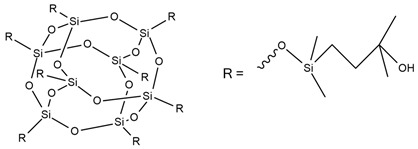	OCTA-POSS	Accelerates the initial stage of degradation, but stabilizes the final decomposition	No data	No data	No data	No data	[40]

## Data Availability

Data are contained within the article.

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
