# Peer review of "A Comprehensive Review of Reactive Flame Retardants for Polyurethane Materials: Current Development and Future Opportunities in an Environmentally Friendly Direction"

_ijms, 2024, doi:10.3390/ijms25105512_

Round 1

Reviewer 1 Report

Comments and Suggestions for Authors

1. In Part 2, the author should focus on the combustion mechanism of polyurethane.

2. In Part 3, is it correct that the carbon nanotubes, expanded graphite, and functionalized graphene oxide described by the author belong to reactive flame retardants?

3. In Part 5, the author should pay attention to the classification and logical combination of the references cited. Here, it is only a simple list and stacking, without in-depth analysis and criticism, and the author's thematic context cannot be seen. There are similar situations in other parts of the article, please make corrections.

4. Table 1 suggests that the author categorize them uniformly according to specific categories. The information in the table is not complete, and the author lists more cases in the previous section than in the table.

5. In Part 6, the author's citation of H. Zhu et al. literature needs to be simplified, highlighting the key points. There are similar situations in other parts of the article, please make corrections.

6. In Part 7, the author should classify and summarize the literature according to phosphorus-nitrogen (P/N), phosphorus-silicon (P/Si), silica-nitro-gen (Si/N), or phosphorus-nitrogen-silica (P/N/Si, and clarify the advantages and disadvantages.

7. In Part 8, the author should indicate the involvement of Silicon containing reactive monomers in the reaction process when citing M. SzoÅ‚yga et al literature.

8. In the perspectives section, is the surface treatment and coating methods mentioned by the author to suppress the release of harmful smoke and improve the flame retardant performance of polyurethane inconsistent with the review topic of reactive flame retardants? The author should describe the current development status and prospects of reactive flame retardants.

9. In the conclusion section, which environmentally friendly flame retardants does the author specifically refer to? Please briefly list and explain the advantages.

Comments on the Quality of English Language

The language should be revised, paying attention to writing and grammar.

Author Response

Dear Editor and Reviewers,

This document responds to reviews of Manuscript with reference ID: ijms-2981508.

Authors of the Manuscript thank you for your valuable comments and suggestions that have helped us improve this article. The work has been thoroughly checked and corrected in all aspects mentioned by the Reviewers. We hope that the improved Manuscript will receive positive feedback.

Yours sincerely,

Paulina Parcheta-Szwindowska, PhD

Reviewer 1:

  1. In Part 2, the author should focus on the combustion mechanism of polyurethane.

Dear Reviewer, thank you very much for this suggestion. Authors included information about the combustion mechanism of polyurethane.

  1. In Part 3, is it correct that the carbon nanotubes, expanded graphite, and functionalized graphene oxide described by the author belong to reactive flame retardants?

Dear Reviewer, thank you very much for the suggestion. The sentence containing the mentioned flame retardants applies to examples of flame retardants from both groups. Nevertheless, we updated information included in Part 3.

  1. In Part 5, the author should pay attention to the classification and logical combination of the references cited. Here, it is only a simple list and stacking, without in-depth analysis and criticism, and the author's thematic context cannot be seen. There are similar situations in other parts of the article, please make corrections.

Dear Reviewer, thank you very much for the suggestion. Unfortunately, we cannot agree with it. the manuscript comprehensive describe various type of reactive flame retardants intended for obtaining various types of polyurethane materials and indicates differences in the flammability of these materials. these differences are described in paragraphs and summarized in tables. This list of cited literature is intended to enable the reader to quickly find the literature that interests him most, whether it concerns PU or TPU flame retardants, the use of silicone flame retardants or others.

  1. Table 1 suggests that the author categorize them uniformly according to specific categories. The information in the table is not complete, and the author lists more cases in the previous section than in the table.

Dear Reviewer, thank you very much for this suggestion. Table 1 shows examples of phosphorous-containing reactive flame retardants and their applications in polyurethane materials. We cannot agree that Part 3 describes more examples than are listed in the table. Each flame retardant example listed in the table was assigned to a specific type of polyurethane material, and more than 1 reference to the literature was added in the last column.

  1. In Part 6, the author's citation of H. Zhu et al. literature needs to be simplified, highlighting the key points. There are similar situations in other parts of the article, please make corrections.

Dear Reviewer, thank you very much for this suggestion. The manuscript has been thoroughly checked and corrected at the mentioned part.

  1. In Part 7, the author should classify and summarize the literature according to phosphorus-nitrogen (P/N), phosphorus-silicon (P/Si), silica-nitro-gen (Si/N), or phosphorus-nitrogen-silica (P/N/Si), and clarify the advantages and disadvantages.

Dear Reviewer, thank you very much for this suggestion. Part 7 was improved by adding literature review according to phosphorus-silicon (P/Si), silica-nitrogen (Si/N), and phosphorus-nitrogen-silica (P/N/Si).

  1. In Part 8, the author should indicate the involvement of Silicon containing reactive monomers in the reaction process when citing M. Szołyga et al literature.

Dear Reviewer, thank you very much for this suggestion. In Part 8, there are included information about the involvement of Silicon containing reactive monomers in the reaction process when citing M. Szołyga et al literature. Please, see page 24, the last paragraph.

  1. In the perspectives section, is the surface treatment and coating methods mentioned by the author to suppress the release of harmful smoke and improve the flame retardant performance of polyurethane inconsistent with the review topic of reactive flame retardants? The author should describe the current development status and prospects of reactive flame retardants.

Chapter titled Perspectives was created due to the desire to present the latest information on various methods of flame retardancy of polyurethane materials. Our goal was to provide various methods currently being developed to reduce the flammability of these materials, including methods not related to the use of reactive flame retardants. These have been comprehensively described in the work.

  1. In the conclusion section, which environmentally friendly flame retardants does the author specifically refer to? Please briefly list and explain the advantages.

Dear Reviewer, information about the prospects for eco-flame retardants has been added in a separate paragraph in the summary section.

Reviewer 2 Report

Comments and Suggestions for Authors

This paper reviews the basics of combustion and current flame retardant compounds used to reduce the flammability of polyurethane foam.  This paper is very thorough and provides a helpful review of current technology.

1.  This reviewer suggests that you review the paper for grammatical errors.  Additionally, make sure all super- and sub-scripts are correctly denoted and all acronyms are defined.  Lastly, ensure all numbers use the same punctuation, as you switch between commas and periods to denote decimals quite frequently.

2. I would expand on the specifics of combustion and pyrolysis of polyurethane foams.  You touch on the depolymerization of hard and soft segments later in the paper, but I think an in-depth look at the degradation process would be helpful at the beginning.  

3.  In lines 64-67, wouldn't phosphorous flame retardants also be listed as a popular reactive compound?

4.  I suggest adding this reference as it showcases a completely bio-based flame resistant polyurethane foam: 

Smith, D.L.; Rodriguez-Melendez, D.; Cotton, S.M.; Quan, Y.; Wang, Q.; Grunlan, J.C. Non-Isocyanate Polyurethane Bio-Foam with Inherent Heat and Fire Resistance. Polymers 202214, 5019. https://doi.org/10.3390/polym14225019

Comments on the Quality of English Language

I suggest reviewing the grammar and making sure the style is consistent (i.e., decimals, defined acronyms, superscripts, subscripts, etc.).

Author Response

Dear Editor and Reviewers,

This document responds to reviews of Manuscript with reference ID: ijms-2981508.

Authors of the Manuscript thank you for your valuable comments and suggestions that have helped us improve this article. The work has been thoroughly checked and corrected in all aspects mentioned by the Reviewers. We hope that the improved Manuscript will receive positive feedback.

Yours sincerely,

Paulina Parcheta-Szwindowska, PhD

Reviewer 2:

  1. This reviewer suggests that you review the paper for grammatical errors. Additionally, make sure all super- and sub-scripts are correctly denoted and all acronyms are defined.  Lastly, ensure all numbers use the same punctuation, as you switch between commas and periods to denote decimals quite frequently.

Dear Reviewer, thank you very much for this suggestion. the manuscript has been thoroughly checked and corrected.

  1. I would expand on the specifics of combustion and pyrolysis of polyurethane foams. You touch on the depolymerization of hard and soft segments later in the paper, but I think an in-depth look at the degradation process would be helpful at the beginning.

Dear Reviewer, thank you very much for this suggestion. Authors included information about the combustion mechanism of polyurethane in Part 4. Polyurethane materials.

  1. In lines 64-67, wouldn't phosphorous flame retardants also be listed as a popular reactive compound?

Dear Reviewer, thank you very much for this suggestion. The manuscript has been thoroughly checked and corrected.

  1. I suggest adding this reference as it showcases a completely bio-based flame resistant polyurethane foam:

Smith, D.L.; Rodriguez-Melendez, D.; Cotton, S.M.; Quan, Y.; Wang, Q.; Grunlan, J.C. Non-Isocyanate Polyurethane Bio-Foam with Inherent Heat and Fire Resistance. Polymers 2022, 14, 5019. https://doi.org/10.3390/polym14225019

            Dear Reviewer, the suggested reference was added.

Comments on the Quality of English Language:

I suggest reviewing the grammar and making sure the style is consistent (i.e., decimals, defined acronyms, superscripts, subscripts, etc.).

Dear Reviewer, thank you very much for this suggestion. The manuscript has been thoroughly checked and corrected.

Round 2

Reviewer 1 Report

Comments and Suggestions for Authors

The authors have revised all questions and recommend acceptance for publication in the current format.

Comments on the Quality of English Language

Grammar and typographical errors in the text should be carefully checked and corrected.